# Multi-Directional Long-Term Recurrent Convolutional Network for Road Situation Recognition

**DOI:** 10.3390/s24144618

**Published:** 2024-07-17

**Authors:** Cyreneo Dofitas, Joon-Min Gil, Yung-Cheol Byun

**Affiliations:** 1Department of Electronic Engineering, Jeju National University, Jeju 63243, Republic of Korea; cdofitas@stu.jejunu.ac.kr; 2Department of Computer Engineering, Jeju National University, Jeju 63243, Republic of Korea; jmgil@jejunu.ac.kr; 3Department of Computer Engineering, Major of Electronic Engineering, Institute of Information Science & Technology, Jeju National University, Jeju 63243, Republic of Korea

**Keywords:** machine learning, deep learning, road situation classification, video classification, convolutional neural network

## Abstract

Understanding road conditions is essential for implementing effective road safety measures and driving solutions. Road situations encompass the day-to-day conditions of roads, including the presence of vehicles and pedestrians. Surveillance cameras strategically placed along streets have been instrumental in monitoring road situations and providing valuable information on pedestrians, moving vehicles, and objects within road environments. However, these video data and information are stored in large volumes, making analysis tedious and time-consuming. Deep learning models are increasingly utilized to monitor vehicles and identify and evaluate road and driving comfort situations. However, the current neural network model requires the recognition of situations using time-series video data. In this paper, we introduced a multi-directional detection model for road situations to uphold high accuracy. Deep learning methods often integrate long short-term memory (LSTM) into long-term recurrent network architectures. This approach effectively combines recurrent neural networks to capture temporal dependencies and convolutional neural networks (CNNs) to extract features from extensive video data. In our proposed method, we form a multi-directional long-term recurrent convolutional network approach with two groups equipped with CNN and two layers of LSTM. Additionally, we compare road situation recognition using convolutional neural networks, long short-term networks, and long-term recurrent convolutional networks. The paper presents a method for detecting and recognizing multi-directional road contexts using a modified LRCN. After balancing the dataset through data augmentation, the number of video files increased, resulting in our model achieving 91% accuracy, a significant improvement from the original dataset.

## 1. Introduction

The rapid advancement in video processing technology has created opportunities to develop beneficial application systems across various fields, such as surveillance and security, medical imaging, autonomous vehicles, entertainment, and gaming, among others. Video processing technology plays a significant role in security and monitoring, especially in road situations, traffic management and legislation, crime and accident prevention, and emergency response [1,2,3]. Video processing technologies enable real-time monitoring and automated detection of road conditions, facilitating the analysis of vast amounts of data collected from road cameras and sensors [4,5,6,7]. Its capability to reconstruct scenarios and sequences of events leading to accidents is significant in formulating improved road safety policies and laws [8]. The road situation is vital in various applications related to autonomous vehicles, intelligent transportation systems, and driver assistance systems, generating a growing volume of data from road sensors, including recognizing other vehicles, pedestrians, traffic signs, and more [9,10]. Roads are vital in various applications related to autonomous vehicles, intelligent transportation systems, and driver assistance systems, generating a growing volume of data from road sensors, including recognizing other vehicles, pedestrians, traffic signs, and more [11,12,13]. To address the challenges associated with road recognition, researchers have explored different approaches, including deep learning and vision-based techniques. With a visual input from a CNN, whose output was fed to the LSTM, this task of road situation recognition serves as a cornerstone for various applications, spanning from video surveillance and behavior analysis to enhancing road safety [14,15]. Researchers have explored different approaches, including deep learning and vision-based techniques, to deal with the challenges associated with road recognition. Recent advancements in this area have demonstrated impressive feats in vehicle demonstrators, such as keeping the car in the correct lane, obstacle avoidance, and even autonomous lane changes [16,17].

This progress, coupled with the growing ubiquity of road cameras, presents a unique opportunity to leverage them for a critical safety concern: the automatic detection and notification of slippery road surfaces. Early detection and awareness of hazardous road conditions are crucial for preventing accidents and ensuring safe driving experiences. Researchers have been exploring the potential of edge computing to extract valuable insights from the road environment and big data [18]. Edge computing processes data closer to its source, enabling real-time analysis and preemptive measures for sustainable and efficient urban transportation, ultimately driving safer and more convenient [19,20]. The device can extract temporal and spatial information from the series of videos captured from the road [21]. One promising approach is the use of multi-directional recurrent neural networks. The multi-directional recurrent convolutional networks for situation recognition are an innovative solution that addresses the complexities and challenges associated with accurately detecting and understanding road situations [22].

The primary objective of road situation recognition is to automatically identify and categorize the activities within a video sequence. A combination of CNN for feature extraction from the input and LSTM for capturing temporal relationships in the data are often used by the majority of road recognition systems. Approaches that use convolutional operators with local receptive fields and computationally demanding classifiers, such as those based on CNN. Since there are quite limited publicly available datasets for road situation classification, constructing a more realistic dataset becomes a crucial task. To address this challenge, a pragmatic approach involves collaboration with industrial partners or relevant organizations that have access to real-world driving scenarios. Improving the automatic recognition results of deep learning models for road situation recognition in practical applications is a multifaceted challenge that requires a comprehensive approach. The current approaches for road situation recognition often require extensive parameterization and weight allocation, which can contribute to increased resource consumption and computational complexity.

The primary objective of road situation recognition is to automatically identify and categorize the activities within a video sequence. Most activity recognition systems traditionally leverage a blend of CNN for extracting features from the data and LRCN to capture the temporal dependencies within the data. Existing CNN-based techniques utilize computationally intensive classifiers and convolution operators with local receptive fields. The primary contributions of this paper are:Scenario detection and interpretation: Our model enhances the ability to accurately detect and interpret diverse road scenarios, despite challenges such as changing lighting conditions, perspective shifts, and environmental factors like shadows. This robustness is crucial for real-time road safety applications.Addressing specific road scenario challenges: Our research tackles the complexities of recognizing various road scenarios that involve variations in lighting, scale, blur, perspective angle, and contrasts within the same classification, as well as similarities between different classes. Advanced convolutional methods are employed to handle these challenges effectively.Comprehensive model comparison: We provide a detailed comparative analysis of multiple models, including CNN2D, CNN3D, LSTM, LRCN, and our proposed model. By evaluating key performance metrics such as accuracy, precision, recall, and F1-score, our study offers valuable insights into the performance of each framework in the context of road scenario recognition.

The remainder of the methodology is structured as follows: Section 2 describes a brief literature study of the offered design framework. Then, we explain the proposed method in Section 3. The implementation process, result, and performance evaluation are in Section 4; finally, we complete this paper in the Section 5 and Section 6.

## 2. Related Studies

This study utilized the LSTM-RNN to identify traffic flow’s spatial and time trends and improve accuracy by using multiple-step prediction [23]. It focuses on identifying and predicting consumer behavior to assist in decision-making, but there needs to be more effort to develop technology that automatically counts customer flow. This study utilizes a non-linear regression model to suggest a recurrent neural network architecture that accurately predicts person counts using low-cost video surveillance recordings. The input videos have color and foreground individual information in an RGBP format. The model brings the advantage of temporal coherence via recurrent layers and spatial data collected through convolutional layers [24]. This study utilized engagement recognition to measure student engagement in online courses using a bidirectional long-term recurrent convolutional network (BiLRCN). The study collected many online instructional videos with students from a typical university. The dataset was constructed using three distinct levels of engagement: low engagement and high engagement. The researchers used learning inputs to annotate the dataset [25]. Several analyses have extended road detection in deep learning, as summarized in Table 1.

Deep learning-based scenario generation was developed to generate training situations automatically for smart sensors and devices in autonomous vehicles. The study used deep learning to extract several occurrences from real road videos, which it then used to replicate in a virtual simulator to create a variety of scenarios. Initially, bounding boxes for each entity in the driving footage were extracted using a Faster Region Convolutional Neural Network (Faster-RCNN). After calculating high-level occurrence bounding boxes, each type of extracted event was classified by an LRCN. A single situation was created by combining the findings of all multiple-event classifications. These created situations can be used to simulate a variety of real-world driving incidents in an autonomous driving simulator [31].

The Vehicle Path Prediction Study is employed to support Advanced Driver Assistance Systems (ADAS), which include a variety of technologies, including Adaptive Cruise Control and Autonomous Braking Systems. This study uses visual data collected by a front vision sensor to anticipate the vehicle’s future course, parameterized as coordinates along the journey. Deep CNNs use recurrent neural networks (RNN) to generate predictions, and they additionally examine the advantages of utilizing recurrence for the task. The models are developed using two different methodologies the RNN and mixture density network [32].

In transportation engineering, significant progress has been achieved in identifying and predicting collisions using video data. By combining multiple data sources, recent studies have used ensemble learning models for recognizing automobile accidents using multimodal data from dashboard cameras, reaching excellent accuracy. Furthermore, further research has concentrated on analyzing and interpreting video data from security cameras using advanced machine learning algorithms, allowing real-time accident situation prediction. These approaches demonstrate how video data may improve traffic management and safety [33].

Present improvements in Internet of Vehicles (IoV)-based vehicle collision detection show innovative methods that combine federated learning (FL) with deep learning. A deep learning approach for model selection in an ensemble learning situation is combined with a knowledge base system to attain near-crash detection without domain-specific knowledge. This approach uses a k-Nearest Neighbors (kNN) algorithm to perform model selection, determines and provides visual features and associated loss values, and trains several models for each image. An intelligent FL approach utilizes global and local model aggregation to group users into clusters to improve data and model security [34].

## 3. Methodology

In this study, we proposed a multi-directional classification of road situations. There are four main steps in the road situation recognition workflow. The initial step is gathering data from surveillance cameras in various road situations. The second phase is creating training, testing, and validation sets for pre-processing the collected data, which consists of data balance, frame extraction, and normalization. The third stage is the multi-directional model, in which a convolutional neural network (CNN) is used for feature extraction of the input. Processing via Long Short-Term Memory (LSTM) layers is then performed to capture temporal relationships. The trained model classifies test data and uses metrics to assess performance in the final stage, which allows video prediction of road situations, as shown in Figure 1.

### 3.1. Dataset

The dataset consists of five classes of road situation videos collected from South Korea’s industry. This project aims to recognize the road situation in a particular environment. The road situation dataset class names are the Driving_reverse, Driving_reverse(Others), Object_falling, Pedestrian, and Stop_vehicle, as shown in Figure 2. The dataset was collected in two different locations during daytime and nighttime. The specific dataset has been chosen for a real-world application requirement. The data consists of videos of road situations from different angles and resolutions. The surveillance camera will detect various kinds of road situations. The road situation captured by the monitoring system will determine the label for later video analysis. It maintains a high level of recognition speed and accuracy. The main drawback is that this technology can only recognize situations in the dataset for training the recognition model. There is a limited amount of publicly available data for road situation datasets. Those datasets also contain mixed categories that are not explicitly related to road situations. We collected 498 videos of the original road situation. We organized the data according to the road situation specifications for the experiment.

Each video in the road situation video dataset contains 10 frames per video. The video’s resolution in our experiment is 1280 × 720 pixels. Other classification datasets contain single objects (cars and falling objects) on the road. A human can easily recognize the video, and it is more accessible to the computer. Other objects in the video can provide additional context. Another issue is that the dataset is handed out as a few videos. The input-only frames from the video are represented as three-dimensional arrays with width, height, and channels, which are pre-processed before being classified into the assigned labels. When starting the research, we classified road situations by assigning them to specific classes.

### 3.2. Preprocessing

During the pre-processing, we needed to feed the datasets to the feature extraction, as real-world data unwanted artifacts. To obtain better outcomes from the data, our model performs the pre-processing of the dataset including the data balancing using data augmentation, extraction of video to frames, resizing, and normalization. The images were processed straight into video frames, a step reproduced in the classification model. During classification, the frame images were resized to 64 × 64 dimensions and normalized. Resizing was required due to the dataset’s uniform frame size of 64 × 64 pixels. To ensure compatibility with the CNN, all images per frame were resized to this specific size. A 3 × 3 color conversion aims to lessen the effect of color variation on image classification, thus streamlining CNN processing. Improving frame quality and system adaptability in road situations are essential. To boost effectiveness throughout network training, image frame normalization included dividing each pixel by the maximum pixel value to ensure uniform distribution of data and improve processing.

#### Extraction of Videos

For better recognition of videos, we extracted 10 frames per video. The videos have a resolution of 1870 × 720 pixels. First, we read the video from the dataset and reformat the video frames to a predetermined length and width. This reformatting step accelerates convergence during the training process. We extracted 10 frames per video, for a total of 15,000 frames from the 750 videos.

Every video has a different situation category and a distinct folder as the classes go. The following section provides the distribution of the dataset:Driving reverse: this class refers to the act of operating a vehicle while moving forward in the opposite lane, with the front end pointing in the direction opposite to the intended travel.Driving reverse (others): this class includes other vehicles misusing the lane, cars parked in the center of the road, people walking in the middle of the road, and people standing in the middle of road.Object falling: this class includes throwing objects from vehicles, people throwing objects onto the road from the sidewalk area, and people in the middle of the road throwing objects toward the center of the road.Pedestrian: this class is a place where people walk in public areas, using their feet rather than a vehicle or other mode of transportation.Stop vehicle: In this class, the cars stopping at any location are captured by the surveillance camera. The driver stepping on the brakes, and the brake lights are enabled.

### 3.3. Feature Extraction

The spatial elements of each of the video frames are preserved by the 2D Convolutional Neural Network (CNN) kernel through a sliding window procedure across the two-dimensional space of the input data. For convolution, the 2D CNN can process a single video frame at a time. Although this assists in identifying things in a single frame, it requires multiple sequential analyses to recognize road situations under different conditions. A convolutional network needs a lot of computing capability to train for every frame separately. Consequently, the temporal features-focused time-distributed approach should be used for sequential video frame analysis. The network effectively keeps the temporal element of the spatial features it extracts by distributing the CNN layers to each time slice of the input using the time distribution function. Max-pooling and convolution layers utilize each other to capture more nonlinear fluctuations in the data. In the CNN part of the model, we used 3 × 3 convolutional kernels at feature extraction, which deepens the network while decreasing the number of parameters and enhancing the model’s capacity for adaptation. (See Table 2).

The processed frame data then enters the feature selection stage, where deep learning techniques are employed to extract relevant features. A CNN2D 2D data analyzes the spatial patterns within each frame, identifying visual cues and objects of interest. To capture the temporal aspects, multiple Long Short-Term Memory (LSTM) layers are utilized, enabling the model to learn and understand the sequential dependencies and changes over time.

It consists of fully connected layers that combine and interpret the spatial and temporal information to make sense of the road situation holistically. The softmax component further refines the predictions, outputting probability scores for different classes or scenarios. The extracted features from the CNN2D and LSTM layers are then fed into the evaluation module, which is the heart of the system.

### 3.4. LSTM Layer

Our multidirectional method harnesses the potential of a Bidirectional Long Short-Term Memory (BiLSTM) network to categorize road scenarios. We employ a bilayer BiLSTM network to extract contextual data relevant to the learning scenario. The BiLSTM, with its potential to enhance the classical LSTM with an multi-inverse operation, enables the network to evaluate objects in pre-learning and post-learning phases. However, the BiLSTM’s potential for real-time video classification is hindered by the increased delay it introduces.

### 3.5. Multi-Directional Long-Term Recurrent Convolutional Network

Our deep learning network model, which incorporates CNN and LSTM networks, is a versatile network capable of processing video sequences, individual image frames, single-value predictions, and sequential predictions. It is a comprehensive architecture for handling various inputs and outputs over time. The current LRCN is used in activity recognition and video recognition. The LSTM, in particular, has played a significant role in improving video classification. By using the forward and backward LSTM, we have demonstrated significant improvements. Our model uses a two-dimensional convolutional neural network (CNN2D) pack with a time-distributed layer for video situation recognition. The video taken in the dataset comprises four main parts: video capture, pre-processing, multi-directional learning, and evaluation layer. We utilized a fully connected layer to process the network outputs and a softmax activation function. These allow the model to provide a probability distribution throughout the range of potential output classes, facilitating accurate predictions. Our study utilizes a two-directional LRCN for video collection where footage from cameras monitoring devices on the roadways is captured. The video data then undergoes pre-processing steps. The individual frames are extracted from the recorded video, representing snapshots of the road scene at different points in time. These extracted frames are prepared for analysis. The resize frames step ensures that all the image data are consistently scaled to a standardized size, allowing for efficient processing. Additionally, feature labels are assigned to each frame, identifying the various elements present, such as vehicles, pedestrians, objects falling, and road markings. The sequence length parameter determines how many consecutive frames will be analyzed together as a sequence, capturing the temporal dynamic of the road situation.

### 3.6. Optimization and Loss Function

During training, the suggested model uses the categorical cross-entropy loss function and the Adam optimizer. Compared to stochastic gradient descent techniques, the popular optimization algorithm known as the Adam optimizer more successfully modifies learning rates to converge to the global minimum. At this point, categorical cross-entropy is the chosen loss function for measuring the difference between the actual and shown class distributions in the classification scheme. This method reveals the proper class with a probability of 1 and the other class with a probability of 0. Our model utilized a softmax classifier in the end layer convolutional network, shown in Equation (Equation 1), utilizing the categorical cross-entropy loss function *L* as shown in Equation (Equation 2). Here, c distinguishes different classes, *p* represents the result probability distribution and $\bar{b}$ stands for the true allocation, described as a one-hot vector.
(1)$$\sigma \left(z_{i}\right)=\frac{e^{z_{i}}}{\sum_{j=1}^{K}e^{z_{j}}}$$
(2)$$L\left(p,\bar{b}\right)=\sum_{c}\left[p_{c}\log\left({\bar{b}}_{c}\right)+\left(1-p_{c}\right)\log\left(1-{\bar{b}}_{c}\right)\right]$$

## 4. Experiments

Our experiments utilized all of the videos from a dataset containing multiple classes. The experimental setup contained data balancing using data augmentation techniques. Performance results were evaluated using a confusion matrix, and we used key performance indicators to assess accuracy, precision, f1-score, and recall. The prediction process for video classification was also thoroughly analyzed.

### 4.1. Experimental Setup

The experimental environment for our model utilizes NVIDIA GeForce RTX 4070 8G (Nvidia, Santa Clara, CA, USA), intel i7-9700 (CPU) (Intel, Santa Clara, CA, USA), and Ubuntu 20.04 (OS), with Keras (a deep learning framework). The input dimension was 20 × 64 × 64 × 3 (NHWC), where 20 represents the number of the input video frame sequence, 64 × 64 defines the image resolution, and 3 defines the three channels of RGB color of frames; the output dimensions are 3-dimensional, and describe its possibility for five classes in road situation classification, respectively, and the dimension in which the maximum value was taken as the final prediction result.

Our research study conducted experiments using 64 × 64-pixel videos consisting of 20 frames each. We explored different combinations of CNN and LSTM layers to evaluate their impact on performance. Then, we divided the dataset into training and test sets with a 75–25% split. We also utilized the Adam optimizer and a categorical cross-entropy loss function to train a model on the training set. We increased the number of videos in the dataset by balancing the data using data augmentation to minimize overfitting, and the model was highly accurate. The model underwent training for 200 epochs with a batch size of 2.

### 4.2. Data Augmentation

To address the overfitting loss, we implemented data augmentation techniques to artificially expand our dataset, as detailed in Table 3. It is important to note that this augmented data may introduce variations compared to additional web or real-life data collected independently [35]. After the initial data collection for each class, we applied four distinct data augmentation methods, including affine transformation, salt, horizontal flip, and linear contrast, to ensure a comprehensive and diverse dataset.

Affine transformations: These are linear transformations that preserve the parallelism of lines. They encompass scaling, translation, and rotation operations [36]. In image frames, the transformation matrices are widely utilized as a convenient tool for performing affine transformations. We use rotation for circular transformation at a certain point or axis. We use the 10 to 20 degree angle of rotation to rotate our video frames, as shown in Equation (Equation 3).
(3)$$\left(\begin{array}{cc} cos_{q} & sin_{q}\\ -sin_{q} & cos_{q} \end{array}\right)$$

Horizontal flip: This augmentation technique involves randomly flipping the input image horizontally (from left to right) with a predefined probability. It is more common than the vertical flip. By mirroring the image along the vertical axis, this method introduces a new perspective and variation to the dataset, potentially enhancing the model’s ability to recognize and generalize patterns across different orientations, as shown in Equation (Equation 4).
(4)$$\left[\begin{array}{c} a^{\prime }\\ b^{\prime }\\ 1 \end{array}\right]=\left[\begin{array}{ccc} -1 & 0 & 0\\ 0 & 1 & 0\\ 0 & 0 & 1 \end{array}\right]*\left[\begin{array}{c} a\\ b\\ 1 \end{array}\right]$$
(5)$$a^{\prime }=-a,b^{\prime }=b$$

Salt: This augmentation technique involves randomly replacing specific pixels with either pure white (salt) or pure black (pepper) values, simulating the effect of random noise or interference in the video data. In this type of noise, the model gains robustness and an enhanced ability to generalize as it learns to recognize patterns, as shown in Equation (Equation 6).
(6)$$p\left(z\right)=\left\{\begin{array}{cc} Pa & forz=a\\ Pb & forz=b\\ 0 & other \end{array}\right.$$

Linear equation: This augmentation method we utilize involves image adjustment. This process involves adjusting the contrast of an image in a linear manner. It can help highlight different features in the image that the model might not learn from the original image, as shown in Equation (Equation 7).
(7)g(i,j)=α.f(i,j)+β

### 4.3. Performance Results on Confusion Matrix

With the model’s predictions in hand, the evaluation phase commences. The accuracy evaluation step assesses the model’s performance by comparing its predictions against ground truth labels, providing insights into its reliability and accuracy. The Confusion Matrix offers a detailed breakdown of the model’s classifications, highlighting areas of strength and potential areas for improvement. We evaluated the classification model’s performance using a confusion matrix. A crucial part of this evaluation was our use of a balanced dataset to choose the classification performance across all models, as shown in Figure 3. For the CNN2D Figure 3a and CNN3D Figure 3b models, out of five classes, four classes were misclassified: seven and three for driving reverse (others), and six and six for object falling, pedestrian, and stop vehicle. The LSTM Figure 3c model misclassified two classes: one instance of object falling and four data points of a stop vehicle. The LRCN Figure 3d model misclassified three classes: one data point in driving reverse, three data points in pedestrian, and seven data points in stop vehicle. Compared to all other models, our model Figure 3e performed most effectively and had the fewest misclassifications. Just four occurrences in the “pedestrian” class and seven in the “stop vehicle” class are misclassified.

### 4.4. Evaluation Metrics

We list the typically utilized vital performance indicators (KPIs) for assessing the effectiveness of classification models: accuracy, precision, recall, and F1-score. The number of correctly predicted target attention is represented by True Positives (TPs), the number of incorrectly predicted target attention is represented by False Positives (FPs), and the number of target attention that was unsuccessfully recognized is represented by False Negatives (FNs). In Equation (Equation 8), accuracy measures how well the model predicts occurrences. It is determined as the ratio of correctly classified samples to the total samples. Equation (Equation 9) defines precision as the percentage of true positives among all positive predictions. Equation (Equation 10) illustrates recall, the rate of true positives among all true positive samples in the collection. Equation (Equation 11) computes the F1-score, the mean of precision and recall that provides even results. We utilize these evaluations during the experiments to determine how our proposed model is evaluated. These measures will make evaluating how accurate the algorithm is at classifying road situations easier.
(8)$$A=\frac{TP+TN}{TP+TN+FP+FN}$$
(9)$$P=\frac{TP}{TP-FP}$$
(10)$$R=\frac{TP}{TP-FN}$$
(11)$$F1-score=2\times \frac{\mathrm{Precision}\times \mathrm{Recall}}{(\mathrm{Precision}+\mathrm{Recall})\;}$$

As Table 4 illustrates, the proposed model outperformed all other models during testing with balanced data in terms of accuracy, precision, recall, and F1-score. In particular, 91%, 89%, 92%, and 91% were achieved by the suggested model. These outcomes indicate a 1% increase in our model’s accuracy comparing the LRCN model, which achieved 90%, 90%, 91%, and 90%. 89%, 88%, 89%, and 89% were generated by the LSTM. Additionally, the suggested model performed significantly better than the CNN3D model, which had results of 74%, 73%, 75%, and 74%. The assessments for the CNN2D model are 71%, 69%, 71%, and 71%, indicating that the suggested model outperformed the other models.

Table 5 shows a comparison of our suggested model’s performance with several current models for road situation detection. In terms of all important performance indicators, our model performs better than the others. In particular, our model’s accuracy, precision, recall, and F1-score are 91%, 89%, 92%, and 91%, respectively. The Faster R-CNN model, on the other hand, is one of the best models for object identification and achieves slightly lower measurements of 89%, 87%, 88%, and 88%. In contrast, the RNN-TCN model, another modern technique, performs well, but still falls short with metrics of 90%, 89%, 88%, and 89%.

The analyzed road situation data can be visualized and utilized in various applications. The model visualization component allows for the interpretation and understanding of the model’s decision-making process, shedding light on the factors it considers most relevant. As shown in Figure 4, to accurately identify each video, we need to make sure that all of the classes are used to recognize it. Through the prediction process, the driving_reverse videos were recognized in 11.76 s, driving_reverse (others) in 11.26 s, pedestrians in 11.24 s, object_falling in 11.29 s, and stop_vehicles in 11.09 s, as shown in Figure 4. Ultimately, the insights gained from this process can enable video prediction capabilities, where the system can anticipate and forecast future road situations based on the observed patterns and dynamics.

## 5. Conclusions

Performing accurate road situation classification is crucial for the safe operation of vehicles. Our proposed study, a multi-directional long-term recurrent convolutional network, improves road situation by promptly and accurately identifying various road situations. This method stands out because it combines a CNN for spatial feature extraction and LSTM networks for capturing temporal dependencies. Key advantages of our method include high accuracy and efficiency, with our model accuracy, precision, recall, and F1-score achieving 91%, 89%, 90% and 91%. The comprehensive classification capabilities allow our model to classify all five road situation classes successfully, outperforming traditional recurrent neural network models. Further, our model shows robust performance by outperforming different models like CNN2D, CNN3D, LSTM, and Faster R-CNN, providing better accuracy and fewer misclassifications in real-world scenarios. We envision a way to recognize road situations in real-time video and its actual system, which needs a more in-depth research approach to recognize different road scenarios.

## 6. Limitation and Future Works

Our study shows promising results in classifying various road situations, but several limitations should be addressed. Although carefully selected, the study area may only partially describe the different road conditions, including varied environments, weather conditions, road types, and unexpected obstacles that can affect model implementation. Some scenarios from the collected dataset, such as pedestrians walking in the same direction as vehicles, seemed unlikely to occur in real-world settings, and were included to test model robustness under various conditions. Future studies should consider more practical scenarios to enhance applicability. Training deep learning models with high traffic volumes presents challenges, including increased computational demands and extensive data preprocessing to handle noise and situations where other objects partially or entirely hide objects in the video footage. Future works could extend the current model to include crash detection and crash risk analysis, providing a more comprehensive approach to road safety and situational awareness. The system could offer predictive insights and suggest preventive measures by integrating features aimed at identifying crash scenarios and assessing factors contributing to crash risks, such as car-following behavior, headway, speeding, and traffic volume. Despite these limitations, our study offers valuable insights into road situation classification and highlights areas for future research to improve model robustness and real-world applicability.

## Figures and Tables

**Figure 1 sensors-24-04618-f001:**
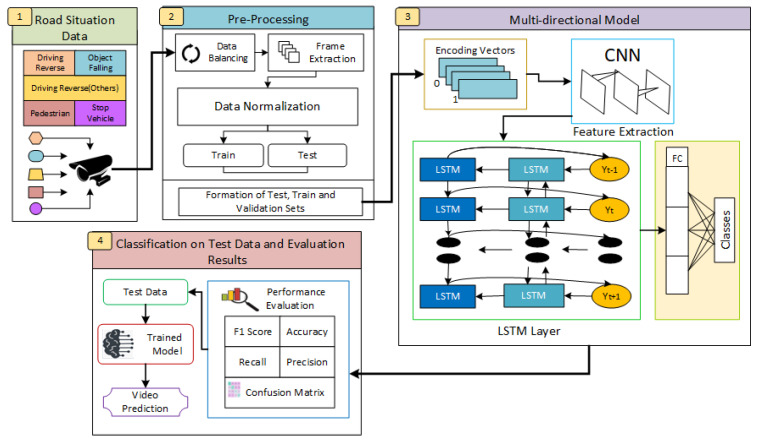
Workflow of the proposed multi-directional LRCN model.

**Figure 2 sensors-24-04618-f002:**
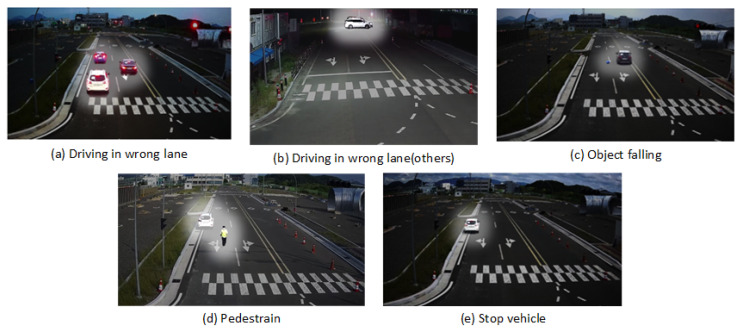
Example of road situation dataset. (**a**) driving_reverse, (**b**) driving_reverse(others), (**c**) object_falling, (**d**) pedestrian, (**e**) stop_vehicle.

**Figure 3 sensors-24-04618-f003:**
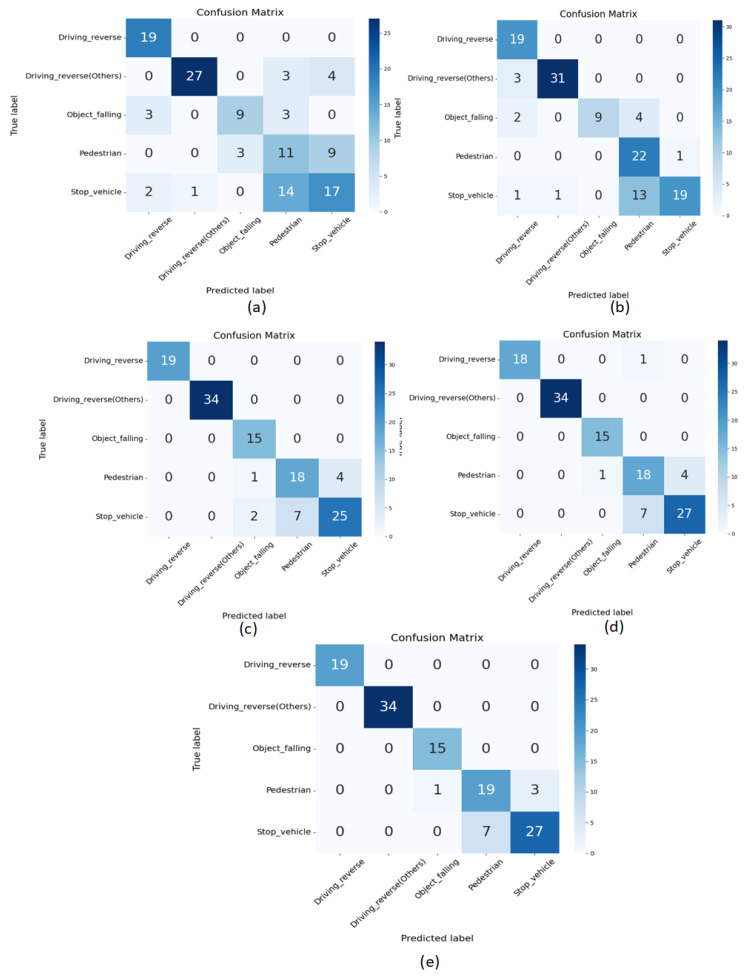
Confusion matrix for experimental results Confusion matrix for experimental results, (**a**) CNN2D, (**b**) CNN3D, (**c**) LSTM, (**d**) LRCN, (**e**) our proposed model.

**Figure 4 sensors-24-04618-f004:**
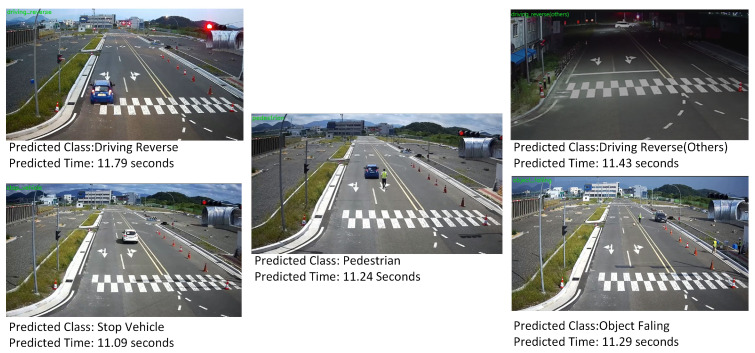
Predicted result using our proposed model.

**Table 1 sensors-24-04618-t001:** Summary of road detection papers in deep learning.

Reference	Year	Model	Dataset	Description
[22]	2020	CNN	Non-public dataset	Works on the determination of the four categories of walking environments (baille blocks, driveways, crosswalks, and sidewalks).
[26]	2020	LSTM-RCNN	Caltech and KITTI traffic	The model that is suggested determines the roadway when a lane is blocked or distorted.
[27]	2020	ANN	Indonesian roads	The method was utilized the technique of recognizing roadblocks, including the ambiguous lines, in static video.
[28]	2021	YOLOv5	Global Road Damage Detection	A random forest model is utilized to detect roadways, trained on features such as the primary color value and the block normalizing position.
[29]	2022	CNN	Automated vehicle location system	Image recognition system for road surface conditions, which can support safety-related decision-making.
[30]	2023	YOLOV3	KITTI	A lightweight model reconstruction and pruning for high-precision. Deployment on mobile devices real-time detection requirements.

**Table 2 sensors-24-04618-t002:** Overview of original dataset.

Class No	Class Name	Videos per Class
0	driving_reverse	100
1	driving_reverse(others)	139
2	object_falling	34
3	pedestrian	111
4	stop_vehicle	114
	Total	498

**Table 3 sensors-24-04618-t003:** Dataset after data augmentation with original test sets.

Class No	Class Name	No. of Videos per Class	Training	Test
0	Driving_Reverse	150	112	19
1	Driving_Reverse(others)	150	112	34
2	Object_Falling	150	112	15
3	Pedestrian	150	112	23
4	Stop_Vehicle	150	112	34
Total		750	560	125

**Table 4 sensors-24-04618-t004:** Comparative analysis of utilizing detection performance on multi-directional LRCN.

Method	Accuracy	Precision	Recall	F1-Score	Training Time min:s
CNN2D	71%	69%	71%	71%	53:16
CNN3D	74%	73%	75%	74%	41:56
LSTM	89%	88%	90%	89%	47:36
LRCN	90%	90%	91%	90%	6:51
Ours	91%	89%	92%	91%	4:13

**Table 5 sensors-24-04618-t005:** Performance analysis of different new models on road situation data.

Models	Accuracy	Precision	Recall	F1-Score
Faster R-CNN	89%	87%	88%	88%
RNN	84%	83%	82%	84%
RNN-TCN	90%	89%	88%	89%
Inception	85%	84%	84%	85%
RCNN	86%	85%	84%	85%
Ours	91%	89%	92%	91%

## Data Availability

Data are contained within the article.

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
