# Peer review of "Multi-Directional Long-Term Recurrent Convolutional Network for Road Situation Recognition"

_sensors, 2024, doi:10.3390/s24144618_

Round 1

Reviewer 1 Report

Comments and Suggestions for Authors

1. The paper provides the basic blocks of the study, but for a general understanding, it would be good to make a general technology pipeline in which all the details would be visible.

2. The results are compared with old models (CNN2D, CNN3D, LSTM, LRCN), today there are many more complex models and it was possible to include something in the comparison.

3. In conclusion, the advantages of the proposed method are discussed. In order to interest the reader (who reads articles from the conclusion :)) you need to briefly give the meaning of your decision and identify which blocks allow you to receive the main preferences.

Reviewer 2 Report

Comments and Suggestions for Authors

This manuscript uses deep learning methods, including Long Short-Term Memory (LSTM) and Convolutional Neural Networks (CNN), to classify five crash situations: driving in reverse, driving in reverse (others), object falling, pedestrian, and stopped vehicle. The data is sourced from surveillance cameras. The paper has merit, as the knowledge gained from this research can be applied to future studies in road safety or further deep learning training. The technical terminology used is appropriate and easy to understand. However, there are several points of concern that need to be addressed and revised in the manuscript, as follows:

  1. There are similar research studies (in the area of using CCTV camera data to analyze crash situations or predict crashes). For example, https://doi.org/10.1016/j.engappai.2024.108350 and https://doi.org/10.1016/j.eswa.2021.115400. The author needs to review papers in the field of Transportation Engineering.
  2. Related to point 1, the author should explain the importance of the road situations studied (why they didn't focus on crash situations or crash risk).
  3. A review of factors related to crash risk should be provided, such as car following behavior, headway, road environment, speeding, or traffic volume.
  4. To evaluate the effectiveness of the classification, a discussion of the situation predictions is needed. For example, in Figure 3 (c-d), there are incorrect predictions for pedestrian and stopped vehicle scenarios. The author should discuss the reasons for these misclassifications.
  5. The limitations of the study should be addressed, such as whether the study area represents real road conditions. What are the differences between the study area and real roads? In Figure 4, some scenarios seem unlikely to occur in the real world, such as pedestrians walking in the same direction as vehicles rather than crossing the street. Additional limitations to consider include: How can deep learning models be trained with high traffic volumes? How can data from CCTV be effectively used for training?

Reviewer 3 Report

Comments and Suggestions for Authors

The manuscript presents a study focusing on road situation classification and demonstrate significant performance enhancements using the proposed method. This research is of substantial value for enhancing the safety of vehicle operations. However, the study does not consider the mixed road conditions in real-world scenarios. Therefore, several essential revisions are required before this manuscript is accepted for publication. Specifically, I recommend incorporating experiments that validate the model under mixed road conditions and enhancing the clarity of certain English expressions as well as the figure legends, including Fig. 3.

Based on these considerations, I suggest accepting the manuscript for publication after minor revisions.

Major comments:

  1. Add comparative experiments on mixed road situation recognition.
  2. Refine the current manuscript with the help of a native English speaker or a professional language editing service.

Minor comments:

  1. In Lines 133-134, the phrase "probably contains" does not clearly express the method. Consider revising for clarity.
  2. For Fig. 3: Add specific method labels in the figure legends to enhance clarity for readers.
  3. For Fig. 3: Improve the clarity of the coordinate labels within the images.
Comments on the Quality of English Language
  1. Refine the current manuscript with the help of a native English speaker or a professional language editing service.

Round 2

Reviewer 2 Report

Comments and Suggestions for Authors

Thank you, my comments have been addressed.